# Wheeled and Seated Mobility Devices Provision: Quantitative Findings and SWOT Thematic Analysis of a Global Occupational Therapist Survey

**DOI:** 10.3390/healthcare11081075

**Published:** 2023-04-10

**Authors:** Hassan Izzeddin Sarsak, Claudia von Zweck, Ritchard Ledgerd

**Affiliations:** 1Occupational Therapy Program, Batterjee Medical College, Jeddah 21442, Saudi Arabia; 2World Federation of Occupational Therapists, 1211 Geneva, Switzerland

**Keywords:** assistive device, education, occupational therapy, wheelchair provision, training, assistive technology

## Abstract

Purpose: To better understand the global role of occupational therapists and explore facilitators and barriers impacting user access to high quality, affordable wheeled and seated mobility device (WSMD) provision worldwide. Methods: Mixed-method approach utilizing quantitative findings and qualitative strengths, weaknesses, opportunities, and threats (SWOT) analysis of a global online survey. Results: A total of 696 occupational therapists from 61 countries completed the survey. Almost 49% had 10 or more years of experience with the provision of WSMDs. WSMD provision had positive, significant associations with attainment of certification (0.000), higher service funding (0.000), higher country income (0.001), standardized training (0.003), continuous professional development (0.004), higher experience (0.004), higher user satisfaction (0.032), custom-made device provision (0.038), higher staff capacity (0.040), and more time working with users (0.050); negative, significant associations were identified with high cost of WSMDs (0.006) and pre-made device provision (0.019). SWOT analysis identified high country income, funding, experience, training, certification from global partners, variety of roles and practice settings, and interdisciplinary teamwork as strengths and opportunities for professional growth, while low country income, lack of time/staff capacity/standardization/support services, and poor access to proper devices were indicated as weaknesses and threats. Conclusion: Occupational therapists are skilled healthcare professionals and provide a variety of WSMD services. Efforts to build collaborative partnerships, enhance access to occupational therapists and funding options, improve service and standards for WMSD service delivery, and promote professional development will help to overcome challenges and facilitate WSMD provision globally. Promoting practices based on best available evidence for WSMD provision worldwide should be prioritized.

## 1. Introduction

### 1.1. Background

Wheeled and seated mobility devices (WSMDs) provide mobility and function for persons with no or restricted ability to ambulate without assistance from technology [1,2]. WSMDs are viewed as among the most important assistive technology devices used in rehabilitation and highly influence the activity of persons with mobility impairments [3,4]. The devices enhance function, improve independence, facilitate the capacity to live at home, and enable community participation [5,6].

Research has shown the positive, significant impact of proper equipment, such as wheelchairs, for the quality of life of individuals with mobility issues, which are defined by the World Health Organization (WHO) as individuals’ perceptions of their position in life in the context of the culture and value systems in which they live and in relation to their goals, expectations, standards, and concerns [7,8]. Properly fitted and correctly prescribed WMSDs benefit both users and their caregivers with the management of daily activities [7]. To obtain such properly fitted devices, however, they must be prescribed by qualified rehabilitation professionals [8]. 

In recognizing the important role of assistive technologies, such as WSMDs, in improving health, the WHO identified that, of the 70 million people worldwide who require wheelchairs, only 5–15% have access to needed products [3]. Issues affecting access to WSMDs include a shortage of health and rehabilitation personnel with the knowledge and skills to provide devices that meet the specific needs of users [7,8,9]. Building the rehabilitation workforce has traditionally not been prioritized in health system funding, with the result that this sector is significantly under-resourced [10]. In addition, health and rehabilitation professionals are not always trained adequately to ensure that people with mobility restrictions receive a quality and custom-fitted wheelchair. There is great variability and inconsistency in whether and how wheeled and seated mobility device-related content is taught and evaluated [11,12]. Furthermore, people in developing countries also often depend on the donation of devices, which are frequently of poor quality and neither suitable nor customized, either for the users or for their environment [13].

### 1.2. Role and Importance of Occupational Therapy in Wheeled and Seated Mobility Device Provision to Global Community

Several health professional groups may be involved in the provision of WSMDs, and they often work together as an interdisciplinary team [13]. Occupational therapists bring a unique perspective grounded in occupation to their approach to WSMD provision. Occupational therapists are health professionals who work with people to facilitate their ability to do the things they want, need, or are expected to do each day [14]. Occupational therapists recognize the value of engagement in meaningful occupations, which are facilitated by appropriately selected WSMDs to achieve health, well-being, and participation in life [15]. By promoting access to appropriate and affordable WSMDs, occupational therapists enable people to become more productive, feel empowered, enjoy leisure, and enhance their participation in their social and cultural environments [14,15].

Assistive technology is an important competency of occupational therapists for optimizing the fit among an individual’s ability and desires to engage in activities, the characteristics of the occupation, and the environment [16]. Occupational therapists may develop their skills in WSMD provision along a continuum from being a generalist to becoming a specialist assistive technology professional [17,18,19].

Individuals throughout the life span with diverse clinical conditions, functional abilities, and mobility goals can benefit from WSMDs provided by occupational therapists [13,14]. To aid in the best quality provision of WSMDs, long-term intervention is required with re-assessment of fit as users age and their functional conditions change [7]. 

Many professional development programs have been developed to enhance the quality of service delivery to WSMD users. For example, standardized training packages were created by a team of experts worldwide by the WHO in partnership with the United States Agency for International Development (USAID) and the International Society of Wheelchair Professionals (ISWP) [9,20,21]. The WHO training packages were developed to serve as a global resource for WSMD advocacy, continuing education, evidence-based practice, innovation, and information exchange. The training promoted eight critical steps in the provision of WSMDs, including: (1) referral; (2) assessment; (3) prescription; (4) funding and ordering; (5) product preparation; (6) fitting and adjusting; (7) user training; and (8) follow-up and maintenance/repairs. Following the identified steps resulted in a range of positive outcomes, including increased satisfaction with devices and better quality of life for the user [11].

The role of occupational therapists in the field of WSMD services has been described in the research literature to include the provision of seating systems or equipment designed to meet the needs of individuals for mobility and postural support and alignment, skin integrity, function, and safety. Such equipment includes positioning systems, mobility devices, durable medical goods, and complex rehabilitation technology used to optimize clients’ environmental access and their ability to safely perform daily activities [19,22].

Research has shown that occupational therapists provide wheeled and seated mobility services across many practice settings, including home health, educational, and rehabilitation facilities; private practice; and community-based settings. Other interdisciplinary team members working with occupational therapists include rehabilitation technology suppliers, technicians, and manufacturers, as well as other healthcare providers and representatives from funding sources [14,23,24].

The World Federation of Occupational Therapists (WFOT) advocates for quality services for people requiring mobility devices, and through collaborative work with global partners, it strives to address barriers faced by occupational therapists in the provision of WSMDs [14]. WFOT endorses the use of guidelines by occupational therapists based on the best available evidence for WSMD provision that supports the occupational therapy process, provides comprehensive services, and integrates a client-centered holistic approach [18,19]. 

As a professional organization representing 107 occupational therapy associations with more than 600,000 occupational therapists around the world, WFOT has sought to gain an understanding of the role of occupational therapy for expanding the capacity for the provision of WSMDs. A global survey was undertaken to: (1) better understand the global role of occupational therapists in the provision of WSMDs; and to (2) gain knowledge regarding facilitators and barriers impacting user access to high quality, affordable WSMDs provided by occupational therapists worldwide. We hypothesized that: (1) occupational therapists play a major role in meeting the goal of improving access to high-quality, affordable WSMDs worldwide; and (2) high quality affordable WSMD provision is positively correlated with factors such as level of country income, availability of assistive technology funding, and service access.

## 2. Method

This study used a two-step mixed method approach to examine the quantitative and qualitative findings of a global online survey. Findings were first calculated for the quantitative results of a global survey. In the second step of the study, the themes arising from the quantitative results were categorized as strengths, weaknesses, opportunities, and threats for WSMD provision using a SWOT analysis approach. The SWOT analysis of the survey findings was undertaken initially by the first author and validated through collaboration with the other authors (see Figure 1).

### 2.1. Data Collection Instrument

To collect data for this study, an online survey was developed by WFOT in English using the SurveyMonkey web tool. The survey was translated by WFOT volunteers to be available in French, German, and Spanish, as well as English. The translated versions of the survey were reviewed by a second translator to ensure accuracy. 

The survey consisted of 22 close-ended questions and required approximately 10 min to be completed. Questions were divided into five sections: (1) demographic information (country income as defined by the World Bank, country of practice, the professional roles of respondents, experience with the provision of WSMDs, practice setting, and service funding); (2) practice description (amount of work time involved with WSMDs, frequency of working with other team members, and time spent working with different age groups, mobility restrictions, and types of WSMDs); (3) role description (assessment of user needs and advice and assistance in obtaining a device) and types of interventions provided (WSMD prescription, delivery, fitting, and follow-up); (4) training in device provision (level and type of training and certification); and (5) quality of service (effectiveness in meeting user needs and user satisfaction with products provided). A four-point rating scale or multiple-choice options were provided to answer questions. Responses were anonymous, and no personal information was collected. The first page of the online survey provided a briefing about the study, and consent was obtained from all participants before starting the survey.

### 2.2. Procedures

The online survey of occupational therapists was conducted by WFOT between November 2021 and January 2022. WFOT invited occupational therapists from all over the world to participate in this survey through a link shared via the WFOT website, WFOT member organizations, social media, and the WFOT e-newsletter. To be eligible for the survey, respondents were required to be occupational therapists who provided services for WMSD users. In accordance with ethical procedures, this study was approved by the Batterjee Medical College Research and Ethical Committee (No.07/05/2022). 

### 2.3. Data Analyses

#### 2.3.1. Step One

Descriptive statistics of simple percentages and frequencies of participants’ responses were calculated to analyze demographic information and other survey results. To measure the linear correlation of two variables of a non-parametric nature, Pearson’s correlation was employed. WSMD provision was evaluated as the dependent variable based on the effect produced by the variations in the independent variables/factors examined in the study. Correlation coefficients and two-tailed significance of the response variables performing the Pearson’s correlation test were used to determine whether responses related to WSMD provision were different across levels of country income, sources of WSMD service funding, cost limitations of WSMDs, standardized training and certification, years of experience, sources of professional development, and time and staff capacity. The data were analyzed using IBM SPSS Statistics software, version 26.

#### 2.3.2. Step Two

Strengths, weaknesses, opportunities, and threats (SWOT) analysis is a fundamental tool for organizations to evaluate their position, and it is widely used to analyze potential internal (strengths and weaknesses) and external (opportunities, threats) factors [25]. In this study, we performed the qualitative SWOT analysis applying a six-phase framework for performing a thematic analysis (become familiar with the data, generate initial codes, search for themes, review themes, define themes, and write up) [25]. The SWOT analysis was undertaken to organize the collected research data and gain insights regarding strengths, weaknesses, opportunities, and threats related to occupational therapy WSMD provision on a global level, as well as to understand potential facilitators and barriers that bear consideration for effectively increasing quality occupational therapy services in the field of WSMDs worldwide.

## 3. Results

### 3.1. Survey Responses

A total of 696 completed individual responses were received from 61 countries (see Figure 2). 

### 3.2. Demographic Information

All respondents were occupational therapists. The majority of respondents lived in high-income countries (86%) and completed the survey in English (64%) or French (21%). Most respondents were occupational therapy practitioners (89%) or educators (20%). Respondents most frequently provided services relating to WSMDs in facilities such as a hospital (47%) or rehabilitation center (44%) or in a personal home (44%). Almost half of respondents had 10 or more years of experience with provision of WSMDs (49%), and an additional 16% had more than five years of experience. Most often, funding for services provided by respondents was public (61%) or mixed public/private (26%).

### 3.3. Practice Description

While almost 30% of respondents were involved for all or most of their working time with WSMDs, most (68%) combined provision of such devices with other occupational therapy interventions. Using a four-point scale to rate frequency between ‘very often’ and ‘never’, on average, respondents indicated that they worked ‘often’ with other team members in the provision of WMSDs, most frequently with wheelchair suppliers, physiotherapists, and other occupational therapists. Respondent scores indicated that occupational therapists work with all age groups but most often with adults and seniors. Unsurprisingly, respondents most frequently worked with people who were completely non-ambulatory or had significant mobility restrictions; WMSD provision was reported only as ‘occasional’ among those able to walk distances of up to 200 m. Manual wheelchairs and walking aids, such as rollators and power wheelchairs, were ‘often’ provided (reported in descending order of average frequency), although electrically powered scooters were reported, on average, to be ‘never’ recommended. Respondents stated that devices were ‘often’ provided completely or partially free of charge to the user; only occasionally were products purchased partially or completely using private funds from the user.

### 3.4. Role Description

Respondents were generally involved in multiple components of the provision of WSMDs. On average, services provided ‘very often’ included assessing the needs of a user and providing advice regarding devices. Providing instructions/training in the use of devices, assisting with applications for obtaining a device, prescribing a device, fitting a device, ensuring device maintenance, and replacing devices were services rated as ‘often’ (in descending order of frequency). Occasional roles included supplying devices, referrals to other health professionals for WSMDs, fundraising for WSMDs, and designing/building devices/device components. 

### 3.5. Training in Device Provision

About 80% of respondents reported a sufficient amount of training for competency in their work in the provision of WSMDs. Most frequently, respondents received training relating to device provision as part of entry level education (65%), through peer-to-peer demonstration and supervised practice (65%), mentoring (55%), and/or other continuing professional education (55%) (see Figure 3). Only 21% of respondents received certification for provision of WSMDs. Professional development needs were most frequently identified related to designing/building or fitting devices, assessing the needs of users, and providing advice to users regarding devices.

### 3.6. Quality of Service

More than 71% of respondents rated user satisfaction with device provision as high or very high, and more than 59% of respondents rated their effectiveness in meeting users’ needs for WMSDs as high or very high, especially among occupational therapists who practice in high-income countries. Limitations of service provision were most often reported as related to lack of staff time/capacity, financial costs to the user and/or health/social system especially in low-income countries, lack of transportation to access WMSDs, travel distances and geography, and stigma associated with accessing WMSDs.

### 3.7. Associations Analysis 

The results of non-parametric correlations of demographics and different variables/factors are presented in Table 1 and confirm our second hypothesis. WSMD provision had a significant, strong, positive correlation with attainment in certification (0.000), availability of public funding for service provision (0.000), higher levels of country income (0.001), involvement in standardized training (0.003), participation in continuous professional development (0.004), and more years of experience (0.004). WSMD provision had a significant, moderate, positive correlation with higher levels of user satisfaction (0.032), higher levels of provision of custom-made devices (0.038), staff capacity (0.040), and greater availability of time spent working with users (0.050). WSMD provision had a significant, strong, negative correlation with higher financial costs to the user (0.006) and a significant, moderate, negative correlation with the provision of pre-made devices (0.019) (see Table 1).

### 3.8. SWOT Thematic Analysis 

Thematic analysis was performed to identify themes related to the quantitative survey results for classification in the strength, weakness, threat, and/or opportunity quadrants of the SWOT framework. SWOT data collected from the occupational therapists were analyzed using a six-step framework [25]. 

#### 3.8.1. Strengths

High country income, increased levels of experience, role competency, training in WSMDs through entry-level education, certification, peer-to-peer demonstration, supervised practice, and mentoring were identified as strengths in this study. Proper customization and high user satisfaction, a wide variety of practice settings in which OTs provide WSMDs, a wide variety of roles assumed by OTs in WSMDs, ability of OTs to combine WSMDs with other interventions, and the ability to provide a wide variety of WSMDs to all age groups with all levels of mobility restrictions were also identified as strengths. 

#### 3.8.2. Weaknesses

Low country income, low availability of work time, lack of staffing capacity, low levels of certification, poor customization and low user satisfaction, lack of standardized screening and referral mechanisms, lack of opportunity for follow-up to ensure appropriate WMSD service, and low support services (lack of financial support, lack of accessible storage areas, lack of infrastructural services) were all identified as weaknesses in this study.

#### 3.8.3. Threats

Financial costs to the user and health systems, lack of public policy for funding support for WMSDs, stigma, and poor access to services/proper devices (travel distance and geography, lack of accessible transportation, and lack of multi-professional services) were all identified as threats in this study.

#### 3.8.4. Opportunities

Affordable WSMDs, availability of public funding, interdisciplinary teamwork, innovative service delivery models, certification from global partners, and the availability of continuing professional development were all identified as opportunities in this study. 

The analysis identified themes at the semantic level. For internal factors, a similar theme for strengths and weaknesses was identified (i.e., related to country income, availability of device funding, occupational therapist certification). For instance, country income is a strength if occupational therapists work in high-income countries but a weakness if they work in low-income countries. A review of external factors indicated a number of themes for opportunities to improve access of occupational therapy WSMD provision by addressing threats to the external support needed for providing WSMDs, including effective service provision models, policies, and funding. Figure 4 provides a summary of the SWOT thematic analysis of occupational therapy WSMD provision as derived from participants’ responses and the themes that had been identified for each SWOT quadrant (see Figure 4).

## 4. Discussion

The purpose of this study was to better understand the role of occupational therapists with provision of WSMDs and to gain knowledge regarding potential facilitators and barriers impacting user access to high quality, affordable WSMDs provided by occupational therapists worldwide. Results of this study confirm our first hypothesis and indicate that occupational therapists work in a wide variety of practice settings in the provision of WSMDs, often combining the use of the devices with other interventions to assist users to improve mobility and participate in needed and wanted activities of daily life. The results support the findings of previous research indicating that occupational therapists have a broad role in most aspects of WSMD provision [14,15,18,20,22].

In this survey, occupational therapists on average reported high levels of competence and user satisfaction in the provision of a variety of types of WSMDs to people of all ages, with all levels of mobility restrictions. Research has found that occupational therapy has proved to be effective in enabling WMSD users to feel empowered, become more productive, enjoy more leisure, and enhance their functional performance in activities of daily living [13,14,20]. Receiving properly fitted WSMDs is known to enhance quality- and health-related outcomes, such as the functional performance of users in terms of independence, safety, and quality; overall experience; and self-perceived satisfaction [13,21,23,26,27].

Occupational therapists are WMSD providers with in-depth knowledge of occupational performance and participation; this knowledge provides a distinct perspective on the impact of habits, roles, and routines regarding the life of an individual to determine the equipment that will be most beneficial in all environments, using a client-centered, participatory approach to interventions to improve functioning and the ability to perform activities that are important to the person [12,14,15].

This study sheds light on different factors that affect occupational therapy WSMD provision as perceived by occupational therapists worldwide. Although the study highlighted the strengths of occupational therapists for effective and competent WSMD service, a lack of time and staffing capacity to fully address user needs was often identified as a weakness. Human resources data collected by WFOT indicate a high level of variation in the per capita supply of occupational therapists for services such as WSMD provision in jurisdictions around the world [28]. Research has indicated that such variations are not related to the population’s need for rehabilitation but instead suggests other factors, such as finances and population size, drive health workforce policy [29,30].

Given the evidence that supports the provision of WMSDs for the health and well-being of people with mobility restrictions [7,8] and the key role of occupational therapy in the delivery of services for such devices, the need is evident for the expansion of the occupational therapist workforce. This identified need is supported by the conclusions of a systematic analysis of the Global Burden of Disease Study 2019 that at least one in every three people worldwide requires rehabilitation at some point, necessitating the provision of services such as occupational therapy in communities as an integral part of primary health care [10].

Findings of the SWOT thematic analysis in this study have actually brought more opportunities for WSMD provision compared to threats. Our SWOT thematic analysis found that higher levels of country income help occupational therapists to deliver a high-quality and affordable WSMD service. This finding corresponds with previous research indicating financial resources available in higher-income countries help to develop and strengthen systems needed for professional wheelchair service provision; conversely, unmet needs for WSMDs were found to be increased in low- and middle-income countries as result of factors such as limited access to quality wheelchairs, less available skilled health personnel, and higher incidence of disabilities [29]. 

In our study, factors such as improved affordability of WSMDs through public or donated funding, interdisciplinary teamwork, access to professional development, and certification from global partners were identified as promising opportunities to promote quality device provision. Research has shown that services provided by certified healthcare professionals enable WSMD use that meets individual posture, mobility, and daily living needs [31]. It is also well-reported in the literature that adequate financial support provides easy access to appropriate WSMD provision. However, for donated devices to be appropriate and effective, they should be prescribed and customized by well-trained professionals to ensure that the device meets user needs and priorities and that service provision is aligned with the customs and context of the local community, as well as international standards [32,33].

Device users worldwide are at high risk for developing secondary health conditions and dying prematurely due to inappropriate WSMD provision by untrained providers [22,28,34]. Different studies have suggested recommendations to overcome such challenges and to ensure the competency of WSMD providers; such recommendations include comprehensive education programs, valid and reliable competency tests, standardized certification through global organizations, and standardized outcome measures [12,24,34,35,36,37,38,39].

In addition to financial costs to the user or health system, threats to WSMD provision identified from survey findings included service barriers to follow-up, transportation, access to support services, role definition, and screening and referral mechanisms. Opportunities exist to improve service delivery through innovative models. For example, telerehabilitation services have proved to be successful, cost effective, convenient, and useful for WSMD provision [40,41,42,43].

Stigma was also identified in this study as a threat. Stigma in many countries represents one of the most pervasive barriers preventing persons with disabilities from accessing equal rights and opportunities. Assistive devices, including WSMD usage, have been associated with stigmatization, as users may face negative attitudes and behaviors from others [44,45]. Supportive policy is needed to promote education and awareness of the benefits of WSMDs. Mainstreaming can make devices less ‘socially visible’ through improved design or by increasing common usage of specific products. For example, the use and acceptance of rollator walkers have significantly increased with aging of the population, particularly in high-income countries [44,46,47].

From the study results, it is evident that a multi-pronged approach is necessary to address the many strengths, weaknesses, opportunities, and threats associated with the provision of WSMDs by occupational therapists. This study has identified a number of priorities to promote service access and address the multiple and often intertwined factors impacting WSMD provision. Such priorities include: (1) provision of WSMD services that are based on best available evidence; (2) identification of innovative service provision models that meet international standards and are relevant to the local context; (3) accessibility to effective professional development to ensure the competency of WSMD providers; (4) increasing the workforce supply of occupational therapists and other rehabilitation professionals needed to meet population needs for the provision of WSMDs; (5) development of collaborative networks to address the needs of users for WSMD service provision; and (6) advocacy to promote the availability of public policy to provide funding, services, and other support, such as education and awareness, needed to effectively provide WSMD services.

### 4.1. Implications for Rehabilitation

This study is the first to explore and provide a detailed description of the current state of wheeled and seated mobility device (WSMD) provision by occupational therapists across the globe.Findings from this study shed light on the role of occupational therapists in provision of WSMDs and promote better understanding of current and future facilitators and barriers impacting user access to high-quality, affordable devices provided by occupational therapists worldwide.Efforts should be prioritized to build collaborative partnerships, develop innovative service models, use best available evidence, advocate for supportive public policy, and promote the availability and accessibility of occupational therapy services and professional development worldwide related to WSMD provision.

### 4.2. Limitations

This study had limitations. Although the survey was available in four languages and promoted to the global occupational therapy community, the number of respondents is a small percentage of the occupational therapists across the world involved in WSMD provision. As with any survey, the respondent sample in this study is not necessarily representative of the global population of occupational therapists, with participants from high-income countries over-represented in the findings. In addition, some demographic information was not collected regarding respondents, for example, age, education, and gender, which may have limited analysis of the data.

## 5. Conclusions

Occupational therapists are skilled healthcare professionals and provide a variety of services for WSMD users and their caregivers. This study was undertaken to address the global challenges in developing accessible and sustainable wheeled and seated mobility provision. Efforts to build international collaborative partnerships, promote practice based on best available evidence, and encourage professional development in the field of WSMD service provision worldwide will help to overcome challenges; lessen the gaps across different experiences, training, and income levels of countries; and enhance understanding of wheeled and seated mobility provision globally. Calling for policy for funding of WMSDs, increasing supply of rehabilitation professionals, developing innovative service models, and promoting the use of assistive technology worldwide should be prioritized. 

## Figures and Tables

**Figure 1 healthcare-11-01075-f001:**
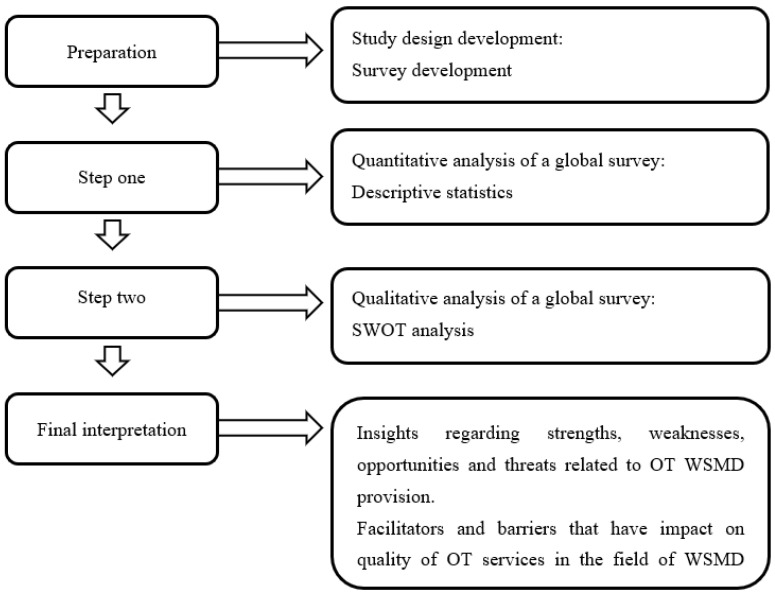
Study design applying the two-step, mixed-methods approach. WSMD: wheeled and seated mobility devices, OTs: occupational therapists, SWOT: strengths, weaknesses, opportunities, and threats.

**Figure 2 healthcare-11-01075-f002:**
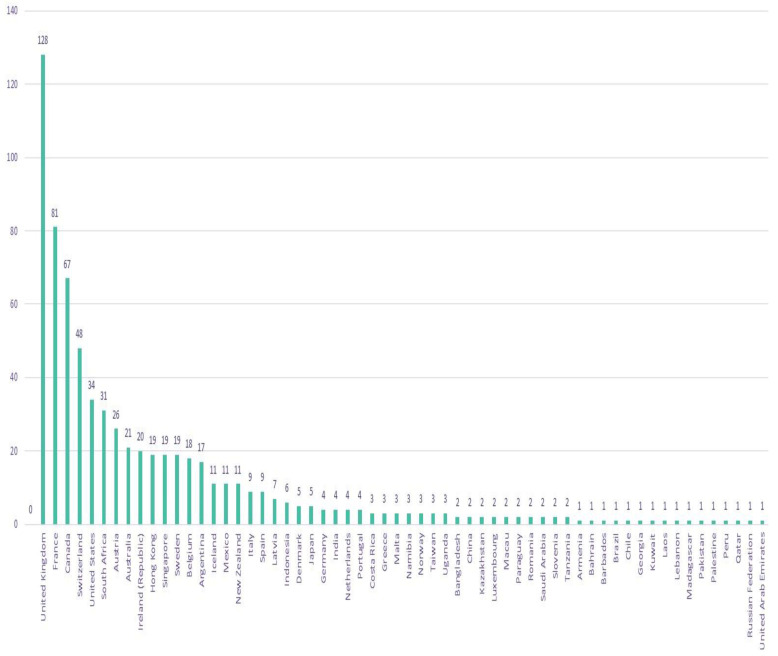
Country of practice of respondents (*n* = 696).

**Figure 3 healthcare-11-01075-f003:**
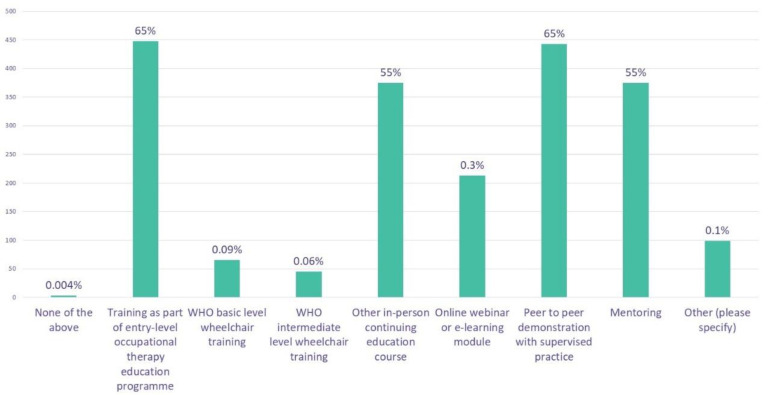
Type of training received specific to wheeled and seated mobility devices.

**Figure 4 healthcare-11-01075-f004:**
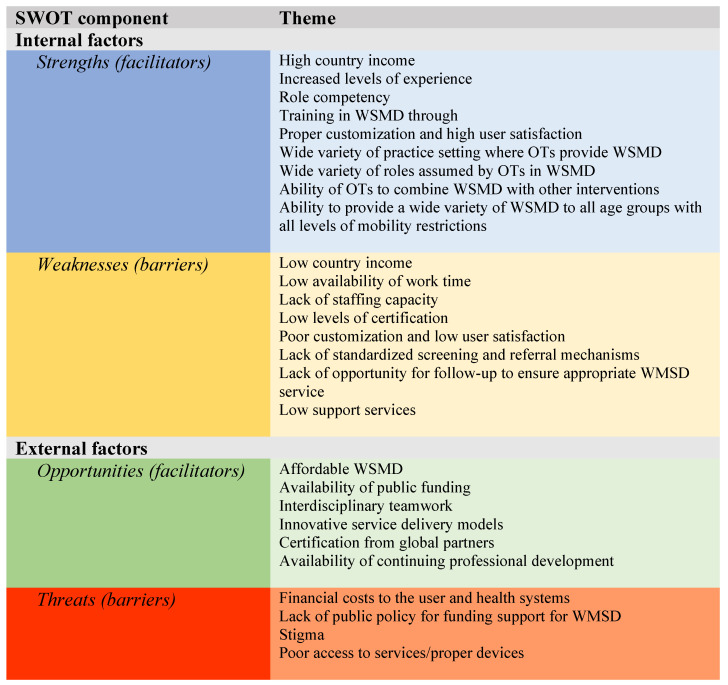
SWOT thematic analysis summary. WSMD: wheeled and seated mobility device, OTs: occupational therapists. Different colors in background are only for a better visual for each category.

**Table 1 healthcare-11-01075-t001:** Significant correlations identified among variables.

	Certification	Public Funding	Income Level of Country	Standardized Training	Continuous Professional Development	Years of OT Experience
WSMD provision	0.840 ** (0.000)	0.716 ** (0.000)	0.743 ** (0.001)	0.851 ** (0.003)	0.851 ** (0.004)	0.785 ** (0.004)
	user satisfaction	provision of custom-made devices	staff capacity	time working with users	cost of WSMD	provision of pre-made devices
WSMD provision	0.511 * (0.032)	0.608 * (0.038)	0.507 * (0.040)	0.602 * (0.050)	−0.843 ** (0.006)	−0.521 * (0.019)

WSMD: wheeled and seated mobility device, OT: occupational therapy. The given values are expressed as R = correlation coefficient (sig—significance–two tailed). Spearman’s correlations, including the nonparametric test, were employed as the statistical test to ascertain the correlations between various variables of the survey. The demographic data are expressed as frequencies with numbers and percentages. ** Highly significant and * significant for the variables.

## Data Availability

Restrictions apply to the availability of these data. Data was obtained from the WFOT and are available on request form the corresponding author with the permission of WFOT.

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
