# Peer review of "Wheeled and Seated Mobility Devices Provision: Quantitative Findings and SWOT Thematic Analysis of a Global Occupational Therapist Survey"

_healthcare, 2023, doi:10.3390/healthcare11081075_

Round 1

Reviewer 1 Report

The poposal of the authors aimed to better understand the global role of occupational therapists and explore facilitators and barriers impacting user access to high quality, affordable wheeled and seated mobility devices (WSMD) provision worldwide.

They proposed amixed method approach utilizing quantitative findings and qualitative strengths, weaknesses, opportunities, and threats (SWOT) analysis of a global online survey. Results: a total of 696 occupational therapists completed the survey from 61 countries.

Their results showed that: (a)  Almost 49% had 10 or more years of experience with provision of WSMD. (b) WSMD provision had positive significant associations with attainment of certification (.000), higher service funding (.000), higher country income (.001), standardized training (.003), continuous professional development (.004), higher experience (.004), higher user satisfaction (.032), custom-made device provision (.038), higher staff capacity (.040), and more time working with users (.050); negative significant associations were identified with high cost of WSMD (.006) and pre-made device provision (.019).

They concluded that (I) occupational therapists are skilled healthcare professionals and provide a variety of WSMD services. (II) Efforts to build collaborative partnerships, enhance access to occupational therapists and funding options, improve service and standards for WMSD service delivery, and promote professional development will help overcome challenges and facilitate WSMD provision globally. (II) Promoting practice based on best available evidence for WSMD provision worldwide should be prioritized.

This is a very interesting piece.

I have some minor suggestions with a pure academic spirit:

1. Eliminate the numbers in the abstracts. They seem to not add much.

2. “Medical College Research and Ethical Committee (No.07/05/2022)” It is a date

4. I suggest to add a flowchart in the methods to better clarify the design of the study (step 1…step 2.. etc.).

5. Improve figure 1.

6. Introduce the themes of the results (arranged in paragraphs) using a few lines

Reviewer 2 Report

This is a carefully executed study with a hight sample size. I appriciate the correctly composed limitations. There are only some formal notices:

1. Figure 1. must be rotated

2. In Table 2. the border between the SWOT categories should be clearer, e.g. skipping one line between two categories

3. The size of the letters are several times changing, e.g. in lines 69-71. 

Reviewer 3 Report

Very meaningful research, good overall flow of information.  I made a few edit recommendations but very well prepared document  

Reviewer 4 Report

This paper presents a mixed method approach utilizing quantitative findings and qualitative strengths, weaknesses, opportunities, and threats (SWOT) analysis of a global online survey    to get better understand the global role of occupational therapists and explore facilitators and barriers impacting user access to high quality, affordable wheeled and seated mobility devices (WSMD) provision worldwide. The idea is interesting and novel. However I would like to give some comments are as follows:

 (1) In the manuscript, once “wheeled and seated mobility device” is defined as WSMD, all subsequent “wheeled and seated mobility device” could be replaced by WSMD for simplicity.

 (2) In line 66-68, why do you think “Occupational therapists recognize the value of engagement in meaningful occupations, which are facilitated by appropriately selected WSMD to achieve health, well-being, and participation in life.”?

 (3) In line 72, “Assistive technology is a core competency of occupational therapists” may be inaccurate.

 (4) I wonder about the data illustrated in Fig.1, many countries only 1 or 2, maybe they cannot be directly compared together.

 (5) Please add a figure to demonstrate the SWOT analysis.

 (6) Please discuss the obtained threshold value with one inter-subject difference source to highlight the obtained criterion's robustness.

 (7) Try to use some experiment result and data to support your method and results.

Reviewer 5 Report

#1: The manuscript is interesting, it is essential that health professionals and specifically the figure of the Occupational Therapy professional provides adequate equipment to improve quality of life, but they talk about quality of life without addressing it conceptually and do not define it. I suggest reviewing and conceptually addressing the term quality of life.

#2: The figures/tables in the manuscript are not clear. Figure 1 shows results from respondents, but is it really necessary? I suggest thinking carefully about the purpose of Figure 1, its importance, clarity and/or whether it is necessary. 

#3: Figure 2 shows specific type of training received, I find it more appropriate to do this in percentages.

#4: Table 2 shows the summary of the SWOT analysis. It provides a lot of information and this sometimes makes it unclear. I suggest modifying it to obtain clearer information.
